# Ibuprofen-Induced Multiple Fixed Drug Eruption Confirmed by Re-Challenge: A Case Report and Literature Review

**DOI:** 10.3390/diagnostics15010048

**Published:** 2024-12-28

**Authors:** Yoshihito Mima, Masako Yamamoto, Hiyo Obikane, Yuta Norimatsu, Ken Iozumi

**Affiliations:** 1Department of Dermatology, Tokyo Metropolitan Police Hospital, Tokyo 164-8541, Japan; 2Department of Diagnostic Pathology, Tokyo Metropolitan Police Hospital, Tokyo 164-8541, Japan; 3Department of Dermatology, International University of Health and Welfare Narita Hospital, Chiba 286-8520, Japan; norimanorima@gmail.com

**Keywords:** fixed drug eruption, multiple fixed drug eruptions, ibuprofen, memory CD8-positive T cells, regulatory T cells, Stevens–Johnson syndrome, toxic epidermal necrolysis

## Abstract

**Background:** Fixed drug eruption (FDE) is a type of drug-induced skin inflammation characterized by the recurrence of lesions in the same region following repeated exposure to the causative drug. FDE typically presents as localized spots or plaques without systemic symptoms; however, it can manifest in other forms, such as blisters and papules. In FDE, effector memory CD8-positive T cells that remain dormant in the basal layer after a previous inflammation are reactivated upon re-exposure to the causative drug, leading to the development of erythema at the same sites. **Case Presentation:** Herein, we report the case of a 23-year-old man who developed ibuprofen-induced multiple FDE. The diagnosis was confirmed by detecting a rash immediately following ibuprofen administration, and histopathological findings were consistent with FDE. Ibuprofen is widely available as an over-the-counter medication, and patients may not always report its use—making the diagnosis of ibuprofen-induced FDE particularly challenging. Approximately 24 h following drug-induced CD8-positive T cell activation, regulatory T cells normally infiltrate the epidermis to suppress inflammation and promote resolution. However, in multiple FDE, CD8-positive T cell activity may outweigh that of regulatory T cells, causing uncontrolled inflammation and leading to the spread of poorly-demarcated lesions that can progress to severe drug reactions such as Stevens–Johnson syndrome (SJS) or toxic epidermal necrolysis (TEN). We reviewed 13 cases of ibuprofen-induced multiple FDE. **Conclusions:** Over-the-counter medications can cause multiple FDEs, and the repeated administration of the causative drug can result in severe reactions such as SJS/TEN. The early diagnosis and strict discontinuation of the causative drugs are therefore crucial.

## 1. Introduction

Fixed drug eruption (FDE) is a type of drug-induced skin inflammation that is characterized by recurrent lesions in the same cutaneous regions following repeated exposure to a causative drug. The rash typically appears at the same locations within 0.5–8 h following drug administration. More than 100 different drugs—including trimethoprim–sulfamethoxazole, naproxen, tetracyclines, and barbiturates—have been reported to cause FDE. FDE typically manifests as well-demarcated, localized spots or plaques without systemic symptoms (e.g., fever) [1].

FDE can present a range of skin manifestations—including generalized, linear, bullous, urticarial, pigmented, non-pigmented, migratory, eczematous, psoriasiform, and pigmentary-persistent erythema-like variants. It can affect any part of the body, including the face, tongue, hands, feet, trunk, limbs, and genitals [2]. Its differential diagnoses include erythema multiforme, Stevens–Johnson syndrome (SJS), contact urticaria, cellulitis, herpes simplex infections, pemphigus vulgaris, bullous pemphigoid, and dermal melanocytosis [3,4,5]. Distinguishing bullous FDE from SJS can be difficult [6]. When typical FDE lesions present with multiple bullous lesions affecting both the skin and mucosa, and these lesions cover at least 10% of the body surface area (BSA) and involve at least three of the six distinct anatomical regions—namely, the head and neck, anterior trunk, back, upper limbs, lower limbs, and genital area—the condition is specifically diagnosed as generalized bullous FDE (GBFDE) [7]. Differentiating GBFDE from other drug-induced blistering disorders, such as SJS and toxic epidermal necrolysis (TEN), is essential due to overlapping clinical features [6].

The presence of fever, visceral involvement, atypical blisters or spots, and a lack of previous episodes with similar lesions may indicate SJS/TEN rather than FDE. The typical interval between drug exposure and symptom onset ranges from minutes to 24 h in FDE and between 1 and 3 weeks in SJS/TEN, which can aid in diagnosis [8]. Histopathological examinations can aid in diagnosis [9]. When FDE occurs in the genital area, it is important to differentiate it from sexually transmitted infections such as herpes simplex. Drug involvement should be considered if symptoms persist despite administering treatments for suspected infections [10,11].

Typically, FDE lesions heal spontaneously within 7–10 days following the discontinuation of the causative agent [8]. However, treatments may involve topical corticosteroids for skin symptoms or oral antihistamines for itching [2]. In certain cases, systemic corticosteroids or cyclosporine may be necessary [12,13]. FDE often leaves residual pigmentation after the acute inflammation has subsided. With recurrent episodes, this pigmentation tends to persist for longer periods. However, no such pigmentary changes occur in non-pigmenting FDE, and complete resolution follows the acute inflammation phase. Abramowitz and Noun first introduced the concept of non-pigmented fixed drug eruption in 1937 [14]. Common causative agents include pseudoephedrine, tetrahydrozoline, piroxicam, thiopental, radiopaque contrast agents (iothalamate), diflunisal, and ephedrine [15]. In non-pigmented FDE, pigmentation typically does not occur; therefore, after the initial lesion resolves, no residual pigmentation is present at the site, which may lead to a longer diagnostic process compared to FDE [16]. In contrast to conventional FDE, non-pigmented FDE is characterized by minimal lymphocyte infiltration into the epidermis and the absence of epidermal necrosis, which likely accounts for the lack of melanin deposition and subsequent pigmentation following the resolution of the rash [17,18].

Ibuprofen, a widely used non-steroidal anti-inflammatory drug (NSAID), is commonly prescribed to treat pain, fever, and inflammation [19]. Similar to other NSAIDs, ibuprofen can cause gastrointestinal side effects such as mucosal erosion, peptic ulcers, and gastrointestinal bleeding [19]. Moreover, it has been reported to increase the frequency of cardiovascular toxicity in patients with preexisting cardiovascular conditions and may lead to nephrotoxic manifestations such as ischemic kidney injury, interstitial nephritis, and nephrotic syndrome [20,21]. Other potential side effects of NSAIDs include central nervous system manifestations such as aseptic meningitis, psychosis, and tinnitus, as well as the triggering of asthma attacks [22].

Several reports have documented adverse skin effects associated with ibuprofen—including contact dermatitis, photosensitivity, angioedema, urticaria, and mucosal lesions [23]. Although ibuprofen-induced FDE is becoming increasingly familiar to dermatologists, cases of multiple FDEs caused by ibuprofen remain rare [24].

CD8-positive T cells include central memory T cells, effector memory T cells, and skin resident memory T cells. Central memory T cells are a subset of memory T cells that reside in lymph nodes and secondary lymphoid organs. Central memory T cells maintain long-term immune memory and, upon re-exposure to an antigen, rapidly proliferate and differentiate into effector cells. They possess high self-renewal capacity and long-term survival potential, enabling them to differentiate into effector memory T cells or effector T cells when encountering the antigen again. Effector memory T cells do not return to lymph nodes but instead circulate through the bloodstream and peripheral tissues. While their proliferative capacity is lower compared to the central memory T cells, they exhibit rapid effector functions, such as cytokine production and cytotoxic activity, initiating immediate immune responses at sites of infection. Skin resident memory T cells are memory T cells that permanently reside in specific tissues, such as the skin, without recirculating through the bloodstream. These cells remain in place long after an infection or inflammation has resolved and can trigger rapid, localized immune responses upon re-infection [25].

The detailed etiology of and mechanisms behind FDE remain unclear. Recent studies have suggested that effector memory T cells present in FDE lesions play a key role in the reactivation of these lesions upon re-exposure to the causative drug and are involved in localized tissue destruction [26]. The causative drug is thought to bind to basal keratinocytes, triggering a localized inflammatory response [27]. This inflammation then upregulates the expression of intercellular adhesion molecule-1 (ICAM-1) via cytokines such as tumor necrosis factor-alpha or interferon-gamma. ICAM-1 expression in these lesions leads to the activation of disease-associated epidermal CD8+ T cells [28]. These then release cytotoxic perforin, granulysin, and interferon-γ and activate the Fas receptor–Fas ligand system—ultimately resulting in localized apoptosis and tissue damage [26,28].

In addition to the role of CD8+ T cells as effector memory cells, skin lesions can be exacerbated by the nonspecific recruitment of CD4+ T cells, CD8+ T cells, and neutrophils, which can further enhance tissue damage [26]. After ~24 h, Foxp3+ regulatory T cells infiltrate the epidermis, resolving the inflammation and improving drug eruptions and subsequent pigmentation [29]. When FDE rashes resolve, CD8+ T cells remain dormant at the dermo–epidermal junction under a memorized condition [28]. The persistence of these effector memory CD8+ T cells in the dermo-epidermal junction after the resolution of inflammation may explain the rapid recurrence of FDE lesions at the same site upon re-exposure to the causative drug [30]. The presence of drug-specific CD8+ T cells in the peripheral blood has been confirmed in non-pigmenting FDE, suggesting that CD8+ T cells are involved in its pathogenesis [31].

Herein, we report a case of ibuprofen-induced multiple FDE that was diagnosed via the clinical observation of lesion recurrence following the re-administration of ibuprofen. We review previous cases of ibuprofen-induced multiple FDEs from the literature.

## 2. Case Presentation

A 23-year-old man with no significant medical history and not taking any regular medications or supplements presented with recurrent episodes of multiple oval-shaped erythematous lesions on his trunk and extremities that had occurred over the preceding year only during bad weather. These lesions would resolve over several days without treatment, leaving behind areas of distinctive pigmentation that would later fade. Since he experienced no systemic symptoms such as fever, he did not visit hospitals and monitored the condition himself. However, the erythematous rash and pigmented lesions progressively expanded with every episode, ultimately prompting him to visit our clinic.

At his first visit, a physical examination revealed multiple scattered dark-brown macules on his trunk and extremities ranging from approximately the size of a thumb to the size of a fist (Figure 1). The BSA of the rash was approximately 20%. He had no itching, pain, or other symptoms apart from the skin lesions. Considering the appearance of the rash, our differential diagnoses included FDE, mastocytosis, and dermal melanocytosis. The patient initially denied the use of any medications; therefore, the diagnosis remained unclear based on his clinical course alone. We opted for continuous observation as he declined further tests such as skin biopsies and blood tests.

One month later, the patient returned with similar erythematous lesions that had progressed to dark-brown macules over several days. At this appointment, he recalled having taken over-the-counter ibuprofen to treat headaches that occurred during bad weather, which he had not mentioned previously. Therefore, we suspected multiple FDE caused by ibuprofen. Because the patient once again declined additional investigations, we continued observation while advising him to avoid ibuprofen. Two weeks later, the patient accidentally took ibuprofen again while experiencing a headache. Four hours later, he developed oval erythematous lesions in the same previously hyperpigmented areas, prompting an immediate return to our clinic (Figure 2a,b).

Laboratory examinations—including complete blood count, eosinophil count (500/μL), C-reactive protein level (0.1 mg/dl), and lactate dehydrogenase level (142 IU/L)—all produced results that fell within normal ranges and did not show any significant signs of inflammation. A histopathological examination of the erythematous lesion revealed liquefaction degeneration of the basal layer, lymphocyte infiltration into the epidermis, and Civatte bodies in the epidermis. Significant perivascular lymphocytic infiltration was observed in the dermis (Figure 3a,b).

The immunostaining of the erythema lesion revealed that the infiltrating lymphocytes in the epidermis were mainly CD8-positive T cells (Figure 4a). Few CD4-positive T cells were observed in the epidermis (Figure 4b). No CD20-positive B cells were observed in the epidermis (Figure 4c).

It was ultimately revealed that each instance of worsening erythematous rash had been consistently preceded by the ingestion of over-the-counter ibuprofen. Combined with the clinical and histopathological findings, he was definitively diagnosed with generalized bullous fixed drug eruption (GBFDE) induced by ibuprofen. He was instructed to avoid ibuprofen. At the time of writing this report, he has since switched to using acetaminophen to treat headaches and has experienced no recurrence of the phenomenon.

## 3. Literature Review

Ibuprofen-induced multiple FDE is a rare occurrence. Only 13 cases of ibuprofen-induced multiple FDE, including GBFDE or non-pigmented FDE, have been documented in the literature [4,6,8,17,23,24,32,33,34,35,36,37] (Table 1). Among the 13 patients comprising these cases, five (38%) were female and eight (62%) were male. Their ages ranged between 7 and 71 years, with an average age of 37.5 years and a median of 40 years—indicating that both younger and older individuals can develop multiple FDE. Ten of the patients (77%) had no significant medical history, one (7.7%) had a history of asthma, one (7.7%) had dysmenorrhea, and two (15.4%) had diabetes and hypertension. Four of the patients (31%) developed multiple FDE within 6 h of ibuprofen intake, five (38%) developed symptoms approximately one day after taking ibuprofen, and one (7.7%) exhibited delayed onset four days after ingesting ibuprofen. Seven of the patients (54%) had experienced similar cutaneous episodes in the past but either had not sought medical attention for these or could not receive precise diagnoses if they did. Twelve of the patients (92%) developed skin lesions on their trunks or extremities, while one (7.7%) predominantly exhibited mucosal lesions involving the oral mucosa and eyelid areas. Five of the patients (38%) had mucosal lesions in areas such as the oral cavity or genital region. Twelve of the patients (92%) presented with erythema or red papules, of whom 11 (92%) experienced accompanying pigmentation. Only one of the patients (8.3%) exhibited non-pigmented FDE, wherein the erythema regressed without pigmentation. Five of the patients (38%) experienced blistering or erosive mucosal lesions, among which one (20%) presented with mucosal lesions only and no skin involvement. With regard to treatment, seven of the patients (54%) experienced spontaneous symptom resolution without the need for topical or oral treatment. Two of the patients (15%) were treated with topical corticosteroids alone. One patient (7.7%) required intravenous cyclosporine, two (15%) required oral corticosteroids, and one (7.7%) required combination therapy with oral cyclosporine and corticosteroids. Eight of the patients (54%) were diagnosed based on their clinical course, four (31%) were diagnosed based on positive patch test results, and one (7.7%) was diagnosed based on a positive drug-induced lymphocyte stimulation test (DLST) result. In our case, although a formal drug challenge test was not performed, a diagnosis of ibuprofen-induced multiple FDE was made based on the patient’s clinical history. This represents the only reported case in the literature wherein the recurrence of multiple FDE immediately following the re-administration of ibuprofen was confirmed by both clinical photographs and histopathological findings.

## 4. Discussion

CD8+ memory T cells, in their resting state in the basal layer following prior inflammation, play a crucial role in developing FDE. When the causative drug is administered, it acts on the keratinocytes in the basal layer and upregulates ICAM-1 expression. This triggers the cytotoxic activity of memory CD8+ T cells in the basal layer, releasing cytotoxic perforin, granulysin, and interferon-gamma—ultimately leading to tissue damage. Approximately 24 h after drug administration, Foxp3+ regulatory T cells infiltrate the epidermis, resolving the CD8+ T cell-mediated inflammation. The CD8+ T cells then become dormant at the dermal–epidermal junction and remain in memory. This localized persistence of CD8+ T cells contributes to the recurrence of rashes at the same site in FDE [26,28,30]. In our case, the majority of the lymphocytes infiltrating into the epidermis were CD8-positive T cells, consistent with the immunohistochemical findings typical of multiple FDE. This suggests that, in our patient as well, CD8-positive T cells were triggered by the repeated ingestion of the causative drug and remained dormant in the basal layer. Upon re-exposure to the offending drug, a significant infiltration of CD8-positive T cells into the epidermis occurred, originating from the basal layer. However, the limitation of our report is that we did not perform additional immunostaining to determine whether the CD8-positive T cells were the central memory T cells, effector memory T cells, or skin resident memory T cells.

Recently, a connection between multiple FDE and SJS/TEN has been suggested [27]. It has been hypothesized that regulatory T cells, which play a significant role in resolving inflammation during FDE, significantly affect the margins of the lesions, limit the spread of the rash, and maintain the sharply demarcated appearance of FDEs [27]. However, if the causative drug is not discontinued, CD8+ T cell activity may dominate over that of regulatory T cells. This causes the lesions to expand, lose their clear boundaries, and potentially progress to SJS or TEN [27]. FDE is triggered by the binding of a drug to basal layer keratinocytes, inducing a localized inflammatory response. This activation leads to effector memory T cells releasing perforin, granulysin, and interferon-γ, causing tissue damage [38]. In contrast, SJS/TEN is a severe drug eruption in which drug-specific CD8-positive T cells react to the causative drug, inducing the apoptosis of epidermal cells through the perforin, granzyme B, and Fas–FasL pathways. Furthermore, granulocyte-mediated cytotoxicity and TNF-α amplify tissue destruction, leading to widespread inflammation that affects mucous membranes and internal organs, resulting in severe systemic symptoms [7].

A key difference between FDE and SJS/TEN lies in the extent of regulatory T cell infiltration into inflamed sites and the relative intensity of inflammation driven by CD8-positive T cells versus regulatory T cells. These factors are thought to determine whether the inflammation resolves spontaneously or progresses to SJS/TEN [27]. Additionally, in SJS/TEN, drug-specific T cell-mediated cytotoxicity, genetic associations with HLA and non-HLA genes, T cell receptor restriction, and cytotoxic mechanisms play crucial roles. These factors contribute to the impairment of regulatory T cell function, distinguishing the pathogenesis of SJS/TEN from that of FDE and other drug eruptions [39].

Recent studies have also suggested a potential correlation between eosinophil count and the severity of drug eruptions [40]. Drago et al. reported that in drug-induced cutaneous reactions, the number of peripheral eosinophils increases as the severity of skin symptoms escalates [40]. Higher peripheral blood eosinophil counts are often associated with more severe clinical presentations, frequently requiring systemic treatment [40]. Additionally, such patients tend to experience longer recovery times compared to those without eosinophilia [40]. In our case, the patient’s peripheral eosinophil count was within normal limits, and eosinophilic infiltration in the dermis was not prominent in the histopathological examination. This may explain the relatively mild severity of the multiple fixed drug eruptions in this patient, which did not necessitate systemic treatment.

In cases of multiple FDE with blisters or erosions, purpura and a positive Nikolsky sign may be observed, making clinical differentiation from SJS/TEN challenging. Cases have been reported wherein topical steroids alone were insufficient to achieve resolution, and lesions or blisters expanded, leading to mucosal involvement that ultimately necessitated the administration of oral corticosteroids or cyclosporine treatments [4,6,8,17,23,24,32,33,34,35,36,37]. Thus, the early diagnosis of multiple FDE is critical to avoid situations wherein CD8+ T cell activation becomes predominant and leads to a disease state resembling SJS/TEN, which requires more advanced treatments such as oral steroids or cyclosporine. In the present case, although no blistering or erosions were observed, the erythema tended to expand with the repeated administration of the causative drug, and immunohistochemical staining revealed a significant infiltration of CD8-positive T cells into both the basal layer and epidermis. Thus, if the diagnosis of multiple FDE had not been made and the causative drug continued to be taken, a further infiltration of CD8-positive T cells could have occurred, potentially overwhelming the regulatory T cells’ ability to control the inflammation. This might have led to the formation of blisters or erosions or progressed to SJS/TEN.

Many women commonly use ibuprofen to treat dysmenorrhea; however, in our literature review, ibuprofen was used for this purpose in only one case [24]. Overall, we observed more cases relevant to this topic that involved men [4,6,8,17,23,24,32,33,34,35,36,37]. Based on our observations, men may tend to take over-the-counter ibuprofen as a pain reliever without visiting a medical institution, which can result in repeated use without receiving medical advice regarding the potential risk of drug eruptions—potentially leading to multiple occurrences. By contrast, women may be more likely to take ibuprofen for dysmenorrhea after consulting a gynecologist or other healthcare provider, where they are more likely to be warned about the risk of drug eruptions or receive early treatment for mild cutaneous symptoms, possibly reducing the risk of multiple occurrences.

Among the 13 prior FDE cases we reviewed, seven patients had experienced previous episodes of multiple FDE but either had not sought medical care or were under observation without a precise FDE diagnosis. Ibuprofen is readily available as an over-the-counter medication, and because multiple FDE rarely causes systemic symptoms, patients may accidentally take repeated doses despite prior warnings, as occurred in our case. The repeated administration of the causative drug in multiple FDEs can lead to severe drug eruptions, such as SJS/TEN; therefore, thorough awareness among healthcare providers and the general public regarding this condition would be beneficial.

In eight of the thirteen similar cases we reviewed, the diagnosis of multiple FDE was based on the timing of drug intake, progression of symptoms, and recurrence of rashes in the same location. In four of the cases, a positive patch test for ibuprofen confirmed the diagnosis of multiple FDE; however, it is important to note that patch tests have a high rate of false negatives; therefore, a negative result does not necessarily rule out the causative drug [41]. In one of the cases, a positive DLST result led to the diagnosis of multiple FDE, but this test has high rates of both false positives and false negatives; therefore, diagnoses should be made based on the overall clinical course, even in the presence of positive DLST results [42].

To make a definitive diagnosis of FDE, controlled administration of the suspected drug at a reduced dose, with the observation of rash recurrence at the same site, represents the ideal approach [2]. However, owing to concerns over progression to SJS/TEN, no oral challenge tests were performed in any of the 13 reviewed cases [4,6,8,17,23,24,32,33,34,35,36,37]. Although a formal oral challenge test was not performed in our current case, the same rash appeared immediately after the patient re-administered the causative drug, and cutaneous histopathological results were consistent with FDE, leading to a definitive diagnosis.

We report a case of ibuprofen-induced multiple FDE and reviewed 13 similar cases previously reported in the literature. Even over-the-counter drugs can cause multiple FDEs, and the repeated administration of the causative drug can lead to severe drug eruptions such as SJS/TEN. Therefore, early FDE diagnosis and the strict discontinuation of the causative drug are crucial to its resolution.

## Figures and Tables

**Figure 1 diagnostics-15-00048-f001:**
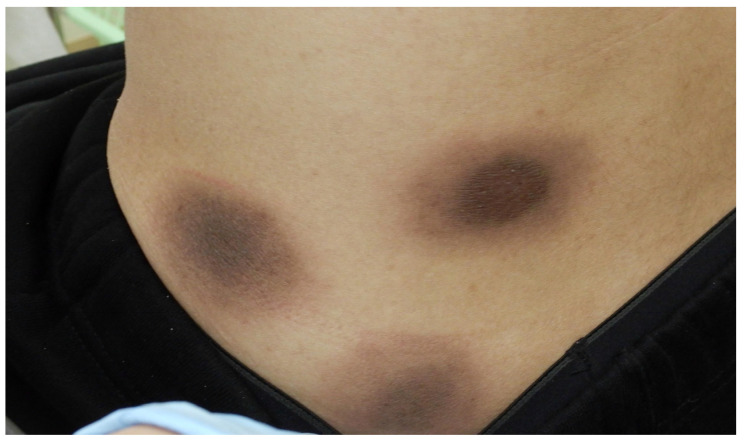
Multiple scattered dark-brown macules on the patient’s trunk and extremities.

**Figure 2 diagnostics-15-00048-f002:**
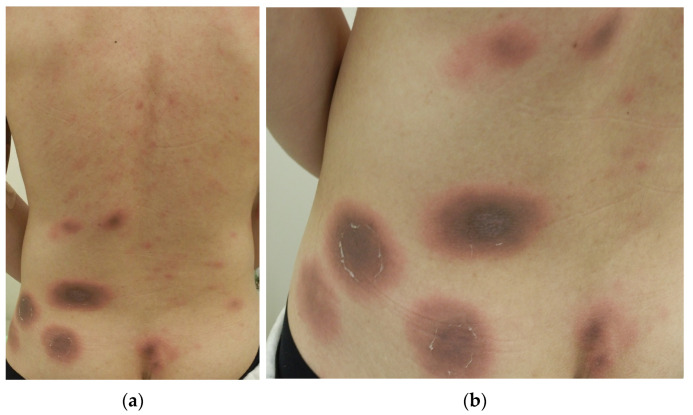
Oval erythematous lesions that developed 4 h after the re-administration of ibuprofen (**a**). A rash emerged at the same previously hyperpigmented areas observable in Figure 1 (**b**).

**Figure 3 diagnostics-15-00048-f003:**
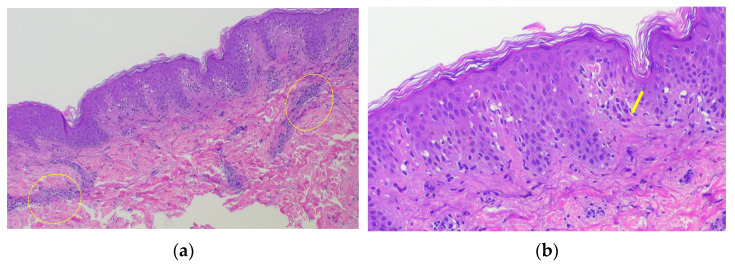
Histopathological examination of the repeated oval erythema revealed significant perivascular lymphocytic infiltration into the dermis (yellow circle) (hematoxylin and eosin [HE] staining; 40× magnification) (**a**). Liquefaction and degeneration of the basal layer, lymphocyte infiltration into the epidermis, and Civatte bodies (yellow arrow) were observed in the epidermis (HE staining; 100× magnification) (**b**).

**Figure 4 diagnostics-15-00048-f004:**
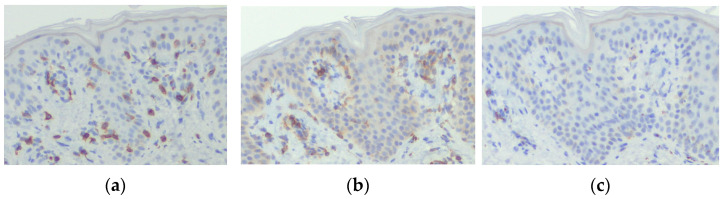
Immunostaining of the erythema lesion shows that the infiltrating lymphocytes are mainly positive for CD8 staining (×100) (**a**), slightly positive for CD4 staining (×100) (**b**), and negative for CD20 staining (×100) (**c**).

**Table 1 diagnostics-15-00048-t001:** The 13 cases of ibuprofen-induced multiple fixed drug eruption previously reported in the literature.

	Age	Sex	Past Medical History	Onset After Drug Intake	Experience of Repeated Rash Emergence	Skin Manifestation	Region of Rash	Treatment	Basis for Diagnosis
Kuligowski ME et al. [32]	49	Female	None	Unknown	None	Edematous erythema	Extremities	None	Patch test
Chen JF et al. [4]	29	Male	None	6 h	None	Dusky red patches with a few overlying blisters and erosions	Trunk, proximal extremities, lip, and foreskin	None	Clinical course
Sánchez-Morillas L et al. [23]	64	Male	None	A few hours	None	Pruritic erythematous macules	Trunk, extremities, and oral mucosa	None	Patch test
Dharamsi FM et al. [8]	30	Male	None	24 h	Yes	Hyperpigmented plaques, some with inlying blisters and erosions	Lips, eyelids, penis, and scrotum	None	Clinical course
Malviya N et al. [33]	40	Male	Asthma	24 h	Yes	Erythematous to violaceous oval macules, coalescing into confluent patches	Trunk, extremities, and periorbital area	Intravenous cyclosporine	Clinical course
Fazeli SA et al. [34]	65	Male	None	24 h	None	Erythematous lesion and oral ulcer	Trunk, extremities, and oral mucosa	Oral corticosteroid and antihistamines, and topical corticosteroids	Clinical course
Singhal RR et al. [17]	71	Female	Diabetes mellitus and hypertension	Unknown	Yes	Erythematous lesions, healed with no pigmentation	Trunk and extremities	None	Clinical course
Tavares Almeida F et al. [35]	47	Female	None	Unknown	None	Annular erythematous plaques with vesicles and pustules	Trunk and extremities	None	Patch test
Bhanja DB et al. [6]	20	Male	None	24 h	Yes	Bullous lesions and erosion	Trunk, extremities, oral mucosa, glans penis, and lips	Oral corticosteroid and antihistamines, and topical corticosteroids	Clinical course
Barootes HC et al. [36]	7	Female	None	4 days	Yes	Erythematous-to-brown patches	Trunk and extremities	Oral corticosteroid and cyclosporine	Drug-induced lymphocyte stimulation test
Shaker G et al. [24]	33	Female	Hypertension, diabetes, and dysmenorrhea	6 h	Yes	Circular patches with a dusky violaceous appearance and surrounding rings of erythema	Trunk and extremities	Topical corticosteroids	Clinical course
Yadav P et al. [37]	10	Male	None	24 h	None	Edematous erythema	Face and trunk	None	Patch test
Our case	23	Male	None	4 h	Yes	Edematous erythema or erythematous patches, some with hyperpigmentation	Trunk and extremities	Topical corticosteroids	Clinical course (reproduction of the rash immediately after taking the causative drug)

## Data Availability

Data concerning this article may be requested from the corresponding author for reasonable purposes.

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
