# Peer review of "Ibuprofen-Induced Multiple Fixed Drug Eruption Confirmed by Re-Challenge: A Case Report and Literature Review"

_diagnostics, 2024, doi:10.3390/diagnostics15010048_

Round 1
Reviewer 1 Report
Comments and Suggestions for Authors
Comments: Authors addressed an important question in the manuscript but manuscript require attention on following comments
1) Line # 89-90; ‘The causative drug is thought to bind to basal keratinocytes, triggering a localized inflammatory response.’ cite appropriate reference to support the statement.
2) Line#91, ‘cytokine 91 pathway’ which cytokine. Author should clearly defined it.
3) Line#98-99; ‘After ~24 h, Foxp3+ regulatory T cells infiltrate the epidermis, resolving the inflammation and improving drug eruptions and subsequent pigmentation.’ appropriate references to be cited to support the statement that Foxp3+ regulatory T cells infiltrate the epidermis within ~24h .
4) Figure 3; Colored arrows are encouraged to define the ‘lymphocyte infiltration and Civatte bodies’ in the epidermis.
5) Line#179-181; sentence can be simplified.
6) Manuscript require proofreading for ‘typos’. e.g. Table 1, last row, ‘Topical corticasoteroids ’
7) Major comment: Authors mentioned involvement of memory CD8+ T cells in both introduction and discussion section of the manuscript and cited references showing the association of same with FDE. Author should clearly define the central memory (TCM), effector memory T cells (TEM) , skin resident memory T cells (TRM). Since authors have not investigated the skin resident memory CD8+ T cells in the biopsy samples, it will mentioned as the limitation of the study. However, immunohistochemistry of resident memory CD8+ T cells using relevant markers is highly encouraged for the improvement of the manuscript.
8) Authors are encouraged to investigate Foxp3+ regulatory T cells (Tregs) in the skin section (if possible), it will support the discussion section where authors mentioned the involvement of Tregs.
9) If possible ration of CD4 and CD8 T cells can be defined in the skin sections. It will support the line#233-234.
Author Response
Comment1:
Authors addressed an important question in the manuscript but manuscript require attention on following comments
1) Line # 89-90; ‘The causative drug is thought to bind to basal keratinocytes, triggering a localized inflammatory response.’ cite appropriate reference to support the statement.
2) Line#91, ‘cytokine 91 pathway’ which cytokine. Author should clearly defined it.
3) Line#98-99; ‘After ~24 h, Foxp3+ regulatory T cells infiltrate the epidermis, resolving the inflammation and improving drug eruptions and subsequent pigmentation.’ appropriate references to be cited to support the statement that Foxp3+ regulatory T cells infiltrate the epidermis within ~24h .
4) Figure 3; Colored arrows are encouraged to define the ‘lymphocyte infiltration and Civatte bodies’ in the epidermis.
5) Line#179-181; sentence can be simplified.
6) Manuscript require proofreading for ‘typos’. e.g. Table 1, last row, ‘Topical corticasoteroids ’
7) Major comment: Authors mentioned involvement of memory CD8+ T cells in both introduction and discussion section of the manuscript and cited references showing the association of same with FDE. Author should clearly define the central memory (TCM), effector memory T cells (TEM) , skin resident memory T cells (TRM). Since authors have not investigated the skin resident memory CD8+ T cells in the biopsy samples, it will mentioned as the limitation of the study. However, immunohistochemistry of resident memory CD8+ T cells using relevant markers is highly encouraged for the improvement of the manuscript.
8) Authors are encouraged to investigate Foxp3+ regulatory T cells (Tregs) in the skin section (if possible), it will support the discussion section where authors mentioned the involvement of Tregs.
9) If possible ration of CD4 and CD8 T cells can be defined in the skin sections. It will support the line#233-234.
Our reply1:
Thank you very much for your detailed and thorough review. We greatly appreciate the numerous comments and suggestions provided to improve this manuscript for publication. Below, we present our responses to the feedback received.
- Thank you for your feedback. We have added a new reference to support the detailed discussion in this section.
- Thank you for your feedback. We have revised the text to specify that the process is mediated by cytokines such as TNF-α and IFN-γ.
- Thank you for your feedback. We have added a new reference to support the detailed discussion in this section.
- Thank you for your feedback. We have indicated the lymphocytic infiltration in the dermis with yellow circles and marked the Civatte bodies with yellow arrows. The figure has been revised accordingly.
- Thank you for your feedback. We simplified the sentences as you suggested.
- Thank you for your feedback. We have corrected the typo in Table 1, as well as any other typos we could identify throughout the manuscript.
- Thank you for your feedback. As you pointed out, there was a lack of detail regarding CD8⁺ T cells, so we have added further information to line 96-109 and 258-261. Additionally, we included a statement acknowledging that a limitation of this study is the lack of staining to distinguish whether the CD8⁺ T cells are effector memory T cells or resident memory T cells.
- Thank you for your feedback. Of course, we also considered staining for regulatory T cells (as some of the cells stained with CD4 are likely to be regulatory T cells). However, there were limitations in performing this analysis at our community hospital.
- Thank you for your feedback. While we considered using digital pathology analysis and flow cytometry to measure the CD4/CD8 ratio, there were limitations in performing these methods at our facility. We apologize for this inconvenience.
Reviewer 2 Report
Comments and Suggestions for Authors
This is a brief review of case reports of fixed drug eruption induced by ibuprofen. The article is well-organized and informative.
1.The title is too long and a shortened title was suggested as below, ibuprofen-induced multiple fixed drug eruption confirmed by re-challenge: a case report and literature review.
2.However, multiple fixed drug eruption is a less commonly used term, and generalized bullous fixed drug eruption (GBFDE) was more commonly used instead.
Whether this case fits the diagnosis of GBFDE remains to be confirmed, and it’s also not clear whether the involvement of BSA is more than 10%.
3.GBFDE and SJS/TEN are distinct diseases with different pathogenic mechanisms, not just the balance between CD8 and regulatory T cells. Please elaborate more about this part in the discussion and incorporate recent studies.
4.Please define more about non-pigmented FDE, and whether there’s difference in challenging dose or individual factors comparing to FDE.
Is there any reported non-pigmented FDE in this review of case series induced by ibuprofen?
Author Response
Comment2:
This is a brief review of case reports of fixed drug eruption induced by ibuprofen. The article is well-organized and informative.
1.The title is too long and a shortened title was suggested as below, ibuprofen-induced multiple fixed drug eruption confirmed by re-challenge: a case report and literature review.
2.However, multiple fixed drug eruption is a less commonly used term, and generalized bullous fixed drug eruption (GBFDE) was more commonly used instead.
Whether this case fits the diagnosis of GBFDE remains to be confirmed, and it’s also not clear whether the involvement of BSA is more than 10%.
3.GBFDE and SJS/TEN are distinct diseases with different pathogenic mechanisms, not just the balance between CD8 and regulatory T cells. Please elaborate more about this part in the discussion and incorporate recent studies.
4.Please define more about non-pigmented FDE, and whether there’s difference in challenging dose or individual factors comparing to FDE.
Is there any reported non-pigmented FDE in this review of case series induced by ibuprofen?
Our reply2:
Thank you very much for your detailed and thorough review. We greatly appreciate the numerous comments and suggestions provided to improve this manuscript for publication. Below, we present our responses to the feedback received.
1.
Thank you for your feedback. We changed our manuscript title as you suggested.
2.
Thank you for your feedback. In this case, the rash covered 20% of the body surface area, with skin erosions present on the trunk and limbs, leading to a diagnosis of GBFDE as you correctly noted. However, since some forms of multiple FDE do not present with bullae or erosions, we have chosen to use the term "multiple FDE" to describe this case. This terminology allows us to consolidate a broader range of cases in our case series. The related information was added to line 51-58 and 145.
3.
Thank you for your feedback. We have added new content referencing a recent literature to highlight that in SJS/TEN, drug-specific T cell-mediated cytotoxicity, genetic associations with HLA and non-HLA genes, T cell receptor restriction, and cytotoxic mechanisms are strongly implicated, as shown in line 268-283. These factors are considered to distinguish SJS/TEN from other drug eruptions, including FDE.
4.
Thank you for your feedback. We have added detailed information on non-pigmented fixed drug eruption (NPFDE) to line 74-82. NPFDE is thought to be induced more frequently by specific drugs such as pseudoephedrine, tetrahydrozoline, and radiocontrast agents; however, we were unable to identify any individual factors contributing to its development. Regarding the dosage used in challenge tests for FDE and NPFDE, no clear differences were found. However, since FDE typically exhibits greater lymphocyte infiltration into the epidermis compared to NPFDE, it is possible that NPFDE may manifest skin symptoms at lower doses than FDE. This suggests that lower doses might be sufficient for oral challenge tests in NPFDE cases. Due to the lack of supporting references, this discussion remains speculative and was not included in the Discussion section. Of the 13 cases reviewed in this report, only one case of non-pigmented FDE due to ibuprofen was reported (Reference 27).
Round 2
Reviewer 2 Report
Comments and Suggestions for Authors
The authors have addressed the questions appropriately, and made necessary supplementation to the aricle.
Author Response
Thank you very much.